# Characterising Exciton Generation in Bulk-Heterojunction Organic Solar Cells

**DOI:** 10.3390/nano11010209

**Published:** 2021-01-15

**Authors:** Kiran Sreedhar Ram, Hooman Mehdizadeh-Rad, David Ompong, Daniel Dodzi Yao Setsoafia, Jai Singh

**Affiliations:** 1College of Engineering, IT and Environment, Purple 12, Charles Darwin University, Darwin, NT 0909, Australia; kiran.sreedharram@cdu.edu.au (K.S.R.); hooman.mehdizadehrad@cdu.edu.au (H.M.-R.); DAVID.OMPONG@cdu.edu.au (D.O.); daniel.setsoafia@students.cdu.edu.au (D.D.Y.S.); 2Energy and Resources Institute, Charles Darwin University, Darwin, NT 0909, Australia

**Keywords:** excitons, organic solar cells, non-fullerene, bulk heterojunction, electric field

## Abstract

In this paper, characterisation of exciton generation is carried out in three bulk-heterojunction organic solar cells (BHJ OSCs)—OSC1: an inverted non-fullerene (NF) BHJ OSC; OSC2: a conventional NF BHJ OSC; and OSC3: a conventional fullerene BHJ OSC. It is found that the overlap of the regions of strong constructive interference of incident and reflected electric fields of electromagnetic waves and those of high photon absorption within the active layer depends on the active layer thickness. An optimal thickness of the active layer can thus be obtained at which this overlap is maximum. We have simulated the rates of total exciton generation and position dependent exciton generation within the active layer as a function of the thicknesses of all the layers in all three OSCs and optimised their structures. Based on our simulated results, the inverted NF BHJ OSC1 is found to have better short circuit current density which may lead to better photovoltaic performance than the other two. It is expected that the results of this paper may provide guidance in fabricating highly efficient and cost effective BHJ OSCs.

## 1. Introduction

High efficiency, stability, low cost and short energy pay back times are the key issues to be considered in the development and commercialization of any type of solar cell to convert solar energy into electricity, which is currently dominated by the silicon solar cells [1,2,3]. The silicon solar cells are highly efficient and stable [4,5] but their fabrication technology is very expensive because it is high temperature based. As a result, much research efforts have currently been devoted to the development of organic solar cells (OSCs) which can be fabricated through the low temperature based chemical process. Among the organic solar cells, the bulk heterojunction (BHJ) OSCs reaching the power conversion efficiency up to 18% [6] have the potential to replace silicon solar cells [7]. The fabrication methods for OSCs are compatible with printing techniques and roll-to-roll processes which are suitable for mass production [8,9]. Currently, research efforts are focused on fabricating highly efficient and stable BHJ OSCs to make them commercially viable and competitive to inorganic crystalline silicon solar cells [10]. As organic solids degrade due to long exposure to sunshine, BHJ OSCs are not only poor in efficiency but also in stability [11]. This opens enormous research opportunities for enhancing the poor efficiency and stability of OSCs which at present are the stumbling blocks towards commercializing this environmentally friendly technology.

The basic structure of a BHJ OSC consists of an active layer composed of two types of blended organic semiconductor materials, a donor and an acceptor [12]. The photogeneration of excitons can occur in both the materials, but when excitons reach a donor-acceptor interface (D-A), the charge transfer (CT) excitons are formed such that the donor donates electrons and accepts holes whereas the acceptor donates holes and accepts electrons [13]. Therefore, donor materials should be efficient hole acceptors and acceptor materials should be efficient electron acceptors [14]. Such an active layer is sandwiched between two electrodes of different work functions that provide an internal electric field in BHJ OSCs to transport the free electrons and holes, generated from the dissociation of CT excitons within the active layer, to their respective electrodes [15].

As organic semiconductors have a low dielectric constant, about 3–4 [16,17,18,19], the electron and hole pairs excited by the absorption of photons by the donor and acceptor in the active layer form Frenkel excitons instantly, which need to be dissociated efficiently into free electrons and holes for the operation of OSCs. Therefore, the research in improving the power conversion efficiency (PCE) focuses on high absorption and fast dissociation of excitons. The dissociation of Frenkel excitons occurs at a D-A interface, which is usually not far away in a blended donor-acceptor active layer. When a Frenkel exciton excited in the donor reaches a D-A interface, the electron gets transferred from the donor’s lowest unoccupied molecular orbital (LUMO) to the acceptor’s LUMO, being at a lower energy, and thus forms a CT exciton [20,21] by releasing the excess energy in the form of molecular vibrations. Likewise, when a Frenkel exciton excited in the acceptor reaches a D-A interface, the hole gets transferred from acceptor highest occupied molecular orbital (HOMO) to donor HOMO, being at a lower energy, and forms a CT exciton in the form of molecular vibrations [13]. If this excess energy is equal or larger than the exciton binding energy and impacts back on the CT exciton, the latter gets dissociated into a free electron and hole which are transported to their respective electrodes by the built-in electric field due to work function difference between the electrodes. After the dissociation, some of the free electrons and holes may recombine before reaching their electrodes and do not contribute to current generation. The recombination losses in OSCs can occur through three major carrier loss pathways, which are the bimolecular, bulk trap assisted, and surface trap assisted pathways [22]. The remaining charge carriers reach their respective electrodes and contribute to the external current. In view of the above, it is clear that the exciton generation plays a crucial role in PCE of BHJ OSCs. It is therefore very desirable to study the profile of exciton generation in OSCs with a view to enhance their photovoltaic performance. 

BHJ OSCs with an active layer based on fullerene acceptor have currently dominated the research activities in organic photovoltaic [23] because of their excellent charge transport properties [24]. Recently, Mehdizadeh Rad et al. [25] applied the optical transfer matrix method (OTMM) to profile the exciton generation rate as a function of the thickness of the active layer for the fullerene acceptor (PCBM) based BHJ OSC in conventional as well inverted configurations. The normalized modulus squared of the electric field and its constructive interference points (CIPs) are analysed using the contour plots. Assuming equal mobility for electrons and holes, and analysing the positions of CIPs within the active layer, they found that for the efficient extraction of charge carriers, excitons should be generated halfway within the active layer; about equal distance from the two electrodes. This is important in the design of organic solar cells to minimize the creation of space charge in the device. These results also conclude that for the same active layer thickness, an inverted fullerene BHJ OSC structure can generate larger short circuit current density than a conventional fullerene BHJ OSC structure [25].

However, fullerene acceptor materials have some disadvantages which include limited chemical and energetic tunability, narrow range of absorption spectra, and unstable morphology, thereby limiting the overall PCE and stability of the device thus fabricated [26,27]. Therefore, the research focus has moved to the use of non-fullerene (NF) acceptors [28,29,30,31] in BHJ OSCs. Recently, Liu et al. [6] fabricated a NF acceptor based BHJ OSC with a PCE of 18.22%, which is the highest to date for a single junction BHJ OSC device. Unlike the fullerene OSCs, the NF devices are much easier to process using solution-based chemistry [32]. It is also possible to increase the device $V_{oc}$ while maintaining a high $J_{sc}$ which is not possible with the fullerene based OSC [33]. This is achieved because of the easy tunability of bandgap energy of the NF acceptor materials and the low binding energy of the CT excitons which reduces the energy loss at the CT state to dissociate the excitons into free charge carriers [33]. To the best of the authors’ knowledge, little theoretical work has been done in understanding the characteristics of exciton generation in conventional and inverted NF acceptor based BHJ OSCs.

Many research activities have been carried out to study the influence of electron transport layer (ETL) and hole transport layer (HTL) in BHJ OSCs [34,35,36]. The work of Rasool et al. [37] aimed at improving the stability of BHJ OSCs by studying the effect of direct contact of the ETL materials on the active layer. Research by Yang et al. [38] focused on improving the built-in potential and reducing the series resistance of BHJ OSCs by optimising the work function of ETL and forming ohmic contacts with the active layer acceptor materials. A report by Lee et al. [39] focused on optimising the thickness of ETL [39]. Most of the researches on HTL are focused on the fabrication of organic and inorganic HTL materials with high transparency and low resistance, as exemplified by the work of Sivakumar et al. [40]. The focus of Chien et al. [41] on HTL was in understanding the inhomogeneity and its effect on the device performance. However, there has not been much focus in understanding the interference effects which the incident light undergoes when passing through ETL and HTL and the effect it will have on the photon absorption and exciton generation in the active layer. 

In view of the above, in this paper, we studied the interference of the incident and reflected electric fields of the electromagnetic radiation and exciton generation rate in the active layer of a conventional fullerene acceptor based BHJ OSC, and two NF acceptor based BHJ OSCs, of inverted and conventional structures, as a function of the thickness of the active layer. In addition, the exciton generation rate was also studied as a function of the thickness of other layers in all three OSCs.

The paper is organized as follows. In Section 1, a detailed introduction is presented, followed by Section 2 which describes the theories that are used to study the interference of incident and reflected electric fields and exciton generation rate in BHJ OSCs, including the study of recombination losses which affect the short circuit current density. In Section 3, the results and discussions are presented, and finally, in Section 4 the conclusions are drawn. 

## 2. Theory

To apply the optical transfer matrix method (OTMM) to study the optical properties, a BHJ OSC can be considered as a multilayer stack as shown in Figure 1. As the thickness of each layer is comparable with the wavelength of incident light, the light absorption and exciton generation are affected by the interference effects [25]. For this work, we coded the OTMM program and used it for simulation on MATLAB version 2020a Update 1 (9.8.0.1359463), USA. 

The total exciton generation rate in the active layer of the BHJ OSC shown in Figure 1 can be written as [25,42,43]:(1)$${\dot{G}}_{j}=\int_{x=0}^{L_{j}}{\dot{G}}_{j}\left(x\right)dx$$
where ${\dot{G}}_{j}\left(x\right)$ is the exciton generation rate at distance x from the top interface of the active layer of thickness $L_{j}$, and it can be given by [25,42,43]:(2)$${\dot{G}}_{j}\left(x\right)=\int_{\lambda =\lambda 1}^{\lambda 2}{\dot{G}}_{j}\left(x,\lambda \right)d\lambda$$
where λ1 and λ2 are the lower and upper wavelength integration limits of solar irradiance, λ is the wavelength of incident solar irradiance and ${\dot{G}}_{j}\left(x,\lambda \right)$ is wavelength and position dependent exciton generation rate given by [25,42,43]:(3)$${\dot{G}}_{j}\left(x,\lambda \right)=\frac{\lambda }{hc}Q_{j}\left(x,\lambda \right)$$
where h is Planck’s constant, c is the speed of light, and $Q_{j}\left(x,\lambda \right)$ is the time average of the energy absorbed per second in the active layer *j* at a position *x* at normal incidence and it is given by [25,42,43]: (4)$$Q_{j}\left(x,\lambda \right)=\alpha_{j}I_{0}T_{int}{\left|t_{j}^{+}\right|}^{2}\frac{n_{j}}{n_{0}}\left\{e^{-\alpha_{j}x}+\rho_{j}^{\prime \prime 2}e^{-\alpha_{j}\left(2L_{j}-x\right)}+2\rho_{j}^{\prime \prime }e^{-\alpha_{j}L_{j}}\cos\left[\frac{4\pi n_{j}\left(L_{j}-x\right)}{\lambda }-\delta_{j}^{\prime \prime }\right]\right\}$$
where $\alpha_{j}=4\pi k_{j}/\lambda$ is the absorption coefficient, $k_{j}$ is the extinction coefficient of the active layer blend, $I_{0}$ is the intensity of incident solar irradiance at the cell surface, $T_{int}$ is the internal transmittance of the substrate, $t_{j}^{+}$ is the internal transfer coefficient which relates the incident plane wave to the internal electric field propagating from the front end to rear end of the cell, $n_{j}$ and $n_{0}$ are the refractive indices of the active layer and substrate, respectively, and $\rho_{j}^{\prime \prime }$ and $\delta_{j}^{\prime \prime }$ are the magnitude and angle of complex reflection coefficient, respectively. The wavelength dependent electric field of the electromagnetic (EM) waves in the active layer *j* at a distance *x* from the top interface is given by [25]:(5)$$E_{j}\left(x,\lambda \right)=\sqrt{\frac{2Q_{j}\left(x,\lambda \right)}{c\varepsilon_{0}\alpha_{j}n_{j}}}$$
where, $\varepsilon_{0}$ is the permittivity of free space. 

Next step is to calculate the short circuit current density, $J_{sc}$, from the total exciton generation rate ${\dot{G}}_{j}$ after considering the recombination losses. The excitons generated in the blend have to reach the D-A interface to undergo dissociation into free electrons and holes [21]. Usually in BHJ OSCs, the exciton diffusion length is less than 10 nm, hence the probability that the excitons diffuse to an interface before recombination is nearly unity [25,44,45,46]. As stated above, the three main recombination loss rates that affect the $J_{sc}$ in a BHJ OSC are the rates of bimolecular recombination loss, ${\dot{G}}_{j}^{bm-rec}$, bulk trap assisted recombination loss, ${\dot{G}}_{j}^{bulk-rec}$, and surface trap assisted recombination loss, ${\dot{G}}_{j}^{surf-rec}$, which are respectively given by [22]:(6)$${\dot{G}}_{j}^{bm-rec}=\frac{eL_{j}}{\varepsilon_{0}\varepsilon_{r}}\xi \left(\mu_{n}+\mu_{p}\right)n^{2}$$
(7)$${\dot{G}}_{j}^{bulk-rec}=\frac{eL_{j}}{\varepsilon_{0}\varepsilon_{r}}\mu_{c}N_{t.bulk}n$$
and
(8)$${\dot{G}}_{j}^{surf-rec}=\frac{e}{\varepsilon_{0}\varepsilon_{r}}\frac{\mu_{s}N_{t.surf}n}{exp\left\{\frac{e\left(V_{b}-V_{cor}\right)}{kT}\right\}}$$
where *n* is the charge carrier density, $\varepsilon_{r}$ is the dielectric constant, ξ is the reduction factor, $\mu_{n}$ and $\mu_{p}$ are the mobility of electrons and holes, respectively, $V_{cor}=V-I_{trans}R_{a}$, is the corrected voltage, V is the applied voltage, $I_{trans}=e\left\{{\dot{G}}_{j}-\left({\dot{G}}_{j}^{bm-rec}+{\dot{G}}_{j}^{bulk-rec}\right)\right\}$ is the transport current density, $R_{a}=\rho L_{j}$ is the series resistance of the active layer, ρ is the resistivity of the active layer blend, $N_{t.bulk}$ and $N_{t.surf}$ are the densities of traps in the bulk and surfaces, respectively, of the active layer, $V_{b}$ is the built-in voltage, $\mu_{c}$ is the mobility of slower moving charge carrier in the bulk, and $\mu_{s}$ is the mobility of the minority charge carrier on the surfaces of the active layer. The net rate of the collected charge carriers reaching their respective electrodes, ${\dot{G}}_{j}^{col}$, can be given by:(9)$${\dot{G}}_{j}^{col}={\dot{G}}_{j}-\left({\dot{G}}_{j}^{bm-rec}+{\dot{G}}_{j}^{bulk-rec}+{\dot{G}}_{j}^{surf-rec}\right)$$
where the rates of recombination losses in Equations (6)–(8) have been subtracted from the rate in Equation (1). Using Equation (9), $J_{sc}$ can be obtained as:(10)$$J_{sc}=e{\dot{G}}_{j}^{col}$$

## 3. Results and Discussions

Simulations of the position dependent exciton generation rate ${\dot{G}}_{j}\left(x\right)$ within the active layer *j* of the following three BHJ OSC1, OSC2, and OSC3 have been carried out. The schematic structures of OSC1, OSC2, and OSC3 are shown in Figure 2a–c, respectively, and the details of their structures are as follows: OSC1: an inverted BHJ OSC with a non-fullerene acceptor of the structure: Glass/indium tin oxide (ITO) (150 nm)/zinc oxide (ZnO) (30 nm)/Poly[(2,6-(4,8-bis(5-(2-ethylhexylthio)-4-fluorothiophen-2-yl)-benzo [1,2-b:4,5-b′]dithiophene))-alt-(5,5-(1′,3′-di-2-thienyl-5′,7′-bis(2-ethylhexyl)benzo[1’,2′-c:4′,5′-c’]dithiophene-4,8-dione)]: C_94_H_78_F_4_N_4_O_2_S_4_ (PBDBTSF:IT4F)/molybdenum trioxide (MoO_3_)(10 nm)/Aluminium (Al) (100 nm) [47,48] (Figure 2a); OSC2: a conventional non-fullerene BHJ OSC of the structure: Glass/ITO(150 nm)/poly(3,4-ethylenedioxythiophene):polystyrenesulfonate (PEDOT:PSS) (30 nm)/PBDB-T-SF:IT-4F/Poly(9,9-bis(3′-(*N*,*N*-dimethyl)-*N*-ethylammoinium-propyl-2,7-fluorene)-alt-2,7-(9,9-dioctylfluorene))dibromide (PFN-Br) (5 nm)/Al (100 nm) [49] (Figure 2b); and OSC3: a conventional fullerene BHJ OSC of the structure: Glass/ITO (180 nm)/PEDOT:PSS (45 nm)/poly(3-hexylthiophene):[6,6]-phenyl C61-butyric acid methyl ester (P3HT:PCBM)/Lithium Fluoride (LiF) (1 nm)/Al (100 nm) [50,51] (Figure 2c). It may be noted that the chosen thicknesses of the layers, other than the active layer, in the above three OSCs are the optimal thicknesses obtained from experiments cited above.

### 3.1. Test of Simulation

Before presenting the details of our simulation, it is desirable to check the accuracy of our simulation by comparing the simulated $J_{sc}$ with some experimental results. For this purpose, we calculated $J_{sc}$ of OSC1, OSC2, and OSC3 using Equations (1)–(10), which require the input parameters given in Table 1 and experimental values of the refractive index $n_{l}$ and extinction coefficient $k_{l}$ (where *l* = 1, 2, *j*, 4, and 5) for PBDBTSF:IT4F and PFN-Br from [49,52], for the Glass, ITO, P3HT:PCBM, PEDOT:PSS, LiF and Al from [50,51,53,54,55], and for ZnO and MoO_3_ from [56]. For completeness, the experimental values of *n_j_* and *k_j_* of the two active layer materials PBDBTSF:IT4F and P3HT:PCBM are plotted in Figure 3 as a function of the wavelength of the incident light.

The simulated $J_{sc}$ for OSC1, OSC2, and OSC3, thus obtained, are plotted as a function of the active layer thickness $L_{j}$ in Figure 4 along with the experimental results from [47] for OSC1, from [49] for OSC2, and from [50,51] for OSC3. According to Figure 4, the simulated results of $J_{sc}$ agree reasonably well with the experimental ones and hence prove that our simulation can be regarded to be reasonably accurate. However, it may be pointed out here that for OSC2, we have the experimental result only for one thickness (see Figure 4).

### 3.2. Electric Field and Exciton Generation Rate Distributions in OSC1

In this section, we present the simulated distributions of the normalised modulus of electric field component $\left|E_{j}\left(x,\lambda \right)\right|$ (Equation (5)) of the electromagnetic (EM) radiation and the subsequent exciton generation rate ${\dot{G}}_{j}\left(x,\lambda \right)$ (Equation (3)) within the active layer of OSC1 using the OTMM method described in Section 2. All input parameters given in Table 1 and $n_{l}$ and $k_{l}$ values for PBDBTSF:IT4F from [49,52], for the Glass, ITO and Al from [50,51,53,54,55], and for ZnO and MoO_3_ from [56] are used in this simulation. Both $\left|E_{j}\left(x,\lambda \right)\right|$ from Equation (5) and ${\dot{G}}_{j}\left(x,\lambda \right)$ from Equation (3) are calculated for three active layer thicknesses—35 nm, 95 nm, and 180 nm—of OSC1 and their contour plots as a function of position in the active layer and wavelength of the solar radiation are shown in Figure 5 in six parts; three on the left-hand side present the electric field distribution in the active layer of three thicknesses—35, 95, and 180 nm—and the corresponding right hand side figures present the distribution of the exciton generation rate.

As a BHJ OSC is a multilayer stack, these brighter and darker regions of $\left|E_{j}\left(x,\lambda \right)\right|$ in the contour plots shown in Figure 5a,c,e are due to constructive interference (CI) and destructive interference (DI), respectively, of the forward travelling electric field, $E_{j}^{+}$, and reflected electric field, $E_{j}^{-}$ of EM waves inside the active layer, as shown in Figure 1. Likewise, in the contour plots of Figure 5b,d,f, the brighter spots represent regions of higher ${\dot{G}}_{j}\left(x,\lambda \right)$ which are also the higher photon absorption regions (HPARs) and lower ${\dot{G}}_{j}\left(x,\lambda \right)$, or lower photon absorption regions (LPARs) are represented by darker spots.

In the active layer, higher generation rate ${\dot{G}}_{j}\left(x,\lambda \right)$, is expected to give higher *J_sc_* and hence an active layer thickness that gives the widest brighter spot for ${\dot{G}}_{j}\left(x,\lambda \right)$ may be regarded to exhibit the optimum performance. However, although all the figures shown in Figure 5 are of the same width, they correspond to different thicknesses of the active layer. That means, widths of the brighter spots in Figure 5a,b are much smaller than those in Figure 5c,d and largest in Figure 5e,f. The wider brighter spot in ${\dot{G}}_{j}\left(x,\lambda \right)$ may occur if the electric field distribution is also wider in the active layer. However, the locations of brighter spots in ${\dot{G}}_{j}\left(x,\lambda \right)$ occur where the wavelength dependent extinction coefficient ($k_{j}$) is non-zero but such spots may not always coincide with the locations of CI of $\left|E_{j}\left(x,\lambda \right)\right|$ for all active layer thicknesses. In view of this, it becomes more important to monitor the brighter spots in ${\dot{G}}_{j}\left(x,\lambda \right)$ whether these spots correlate or not with those in $\left|E_{j}\left(x,\lambda \right)\right|$. However, an optimal thickness of the active layer can still be expected to be in which both ${\dot{G}}_{j}\left(x,\lambda \right)$ and $\left|E_{j}\left(x,\lambda \right)\right|$ have maximum overlap of bright spots. With this view, it may be concluded that the active layer thickness of 95 nm with $\left|E_{j}\left(x,\lambda \right)\right|$ and ${\dot{G}}_{j}\left(x,\lambda \right)$ shown in Figure 5c,d, respectively, is expected to give the optimal photovoltaic performance. This conclusion that the optimal thickness is 95 nm agrees very well with the experimental results of Zhao et al. [47], who found optimal PCE at the same thickness. However, if one compares *J_sc_* shown in Figure 4, the active layer thicknesses of 180 nm has only slightly less *J_sc_* (≈0.03 mA cm^−^^2^ less) than at the optimal active layer thickness of 95 nm. This is because, as explained above, the amount of exciton generation depends on the thickness; with larger thickness more exciton generation rate and hence an active layer of thickness 180 nm is expected to have the widest brighter exciton generation spots leading to *J_sc_* comparable with that of thickness 95 nm. However, at *L_j_* = 180 nm the fill factor (*FF*) is less than that at *L_j_* = 95 nm and hence the PCE reduces. In addition to lower PCE, an increased thickness of $L_{j}=$ 180 nm is expected to require nearly twice the amount of active layer material and therefore is uneconomical.

### 3.3. Influence of Other Layers on Exciton Generation Rate

Here we study the effect of varying the thickness of electrodes and charge transport layers on ${\dot{G}}_{j}$ and ${\dot{G}}_{j}\left(x\right)$ in all three OSCs, which have layer 1 as the front electrode (FE) of thickness *L*_1_, layer 2 as the front charge transport layer (FCTL) of thickness *L*_2_, layer 4 as the rear charge transport layer (RCTL) of thickness *L*_4_, and layer 5 as the rear electrode (RE) of thickness *L*_5_. To the best of the authors’ knowledge, a theoretical study about the influence of thickness of layers other than the active layer on ${\dot{G}}_{j}$ and ${\dot{G}}_{j}\left(x\right)$ in NF BHJ OSCs has not been done yet. For OSC1, using $n_{l}$ and $k_{l}$ values for each layer as described above and other required parameters from Table 1, ${\dot{G}}_{j}$ (Equation (1)) is calculated for two active layer thickness *L_j_* = 95 nm and 180 nm by varying the thicknesses of other layers. ${\dot{G}}_{j}\left(x\right)$ (Equation (2)) as a function of *x* is calculated only for the optimal active layer thicknesses *L_j_* = 95 nm by varying other layer thicknesses. Thus, calculated ${\dot{G}}_{j}$ is plotted as a function of the thickness of each of the four layers on the left column of Figure 6 and the corresponding contour plot ${\dot{G}}_{j}\left(x\right)$ within the active layer of optimal thickness on the right column of Figure 6.

According to Figure 6a,c,e, ${\dot{G}}_{j}$ exhibits an oscillatory behaviour when the thickness of the ITO, ZnO, and MoO_3_ layers are varied. These oscillations may be attributed to the interference effect of the forward and reflected electric fields of the electromagnetic waves. However, such oscillations do not occur when the thickness of Al layer (layer 5) is varied as shown in Figure 6g, which will be discussed later.

In Figure 6a, the maxima of ${\dot{G}}_{j}$ are observed for the ITO thickness *L*_1_ = 148 nm and 19 nm corresponding to the active layer thickness *L_j_* = 95 nm and 180 nm, respectively. From this, it is evident that for thicker active layer *L_j_*, thinner ITO layer L_1_ is required to get maximum ${\dot{G}}_{j}$. Although ${\dot{G}}_{j}$ is the highest for *L*_1_ = 19 nm at *L_j_* = 180 nm, such an OSC may have very large resistive power loss due to the large series resistance offered by the thin ITO layer *L*_1_ [67]. Thus, the simulated optimal thickness of ITO is chosen to be *L*_1_ = 148 nm when the active layer thickness *L_j_* = 95 nm, which is in good agreement with the experimental result [47]. Figure 6b shows the effect of varying *L*_1_ on ${\dot{G}}_{j}\left(x\right)$ at different positions inside the active layer of the optimal thickness, *L_j_* = 95 nm. According to Figure 6b, the most exciton generation rate ${\dot{G}}_{j}\left(x\right)$ occurs within the first 60 nm of the optimal active layer thickness *L_j_* = 95 nm with two maxima (brightest) spots at *L*_1_ = 50 nm and 148 nm, which are consistent with the maxima in ${\dot{G}}_{j}$ shown in Figure 6a.

The effect of varying the thickness *L*_2_ (ZnO) on ${\dot{G}}_{j}$ and ${\dot{G}}_{j}\left(x\right)$ are shown in Figure 6c,d, respectively. It is found from Figure 6c that ${\dot{G}}_{j}$ is high for 0 < *L*_2_ < 30 nm at the optimum active layer thickness *L_j_* = 95 nm, but for the active layer thickness of *L_j_* = 180 nm, ${\dot{G}}_{j}$ is high for 120 < *L*_2_ < 140 nm. The FCTL (ZnO) helps in reducing the charge recombination and contact resistance between the active layer and FE (ITO). Therefore, on one hand, *L*_2_ should be non-zero to serve the above two purposes. On the other hand, all the layers above the active layer should be as thin as possible to maximise the transmission of light to the active layer. For the optimal active layer thickness of *L_j_* = 95 nm, the *L*_2_ may be optimised to maximise photon absorption and least charge recombination loss. Hence, the optimum value of *L*_2_ may be considered to be 30 nm which is in good agreement with the experimental result [47]. According to Figure 6d, the high ${\dot{G}}_{j}\left(x\right)$ occurs within the active layer for 0 < *x* < 40 nm and for 0 < *L*_2_ < 30, which is consistent with the corresponding ${\dot{G}}_{j}$ in Figure 6c.

Figure 6e,f shows ${\dot{G}}_{j}$ and ${\dot{G}}_{j}\left(x\right)$, respectively, as a function of the thickness *L*_4_ of layer 4 (MoO_3_). Here again, the oscillatory nature of ${\dot{G}}_{j}$ occurs due to interference but the amplitude of oscillations is relatively larger in comparison with all other layers. This may be attributed to the fact that the unabsorbed light from the active layer and reflected from the Al cathode passes through this layer again before getting absorbed in the active layer, leading to a larger interference effect. The amplitudes of oscillations for the active layer thickness *L_j_* = 95 nm are much larger than those for *L_j_* = 180 nm. According to Figure 6e, the simulated optimal thickness of *L*_4_ (MoO_3_) is 10 nm at the optimal active layer thickness *L_j_* = 95 nm, which is in good agreement with experimental *L*_4_ [47]. This is also in agreement with ${\dot{G}}_{j}\left(x\right),$ shown in Figure 6f, where the widest bright spot is found to occur at *L*_4_ = 10 nm.

In Figure 6g,h, ${\dot{G}}_{j}$ and ${\dot{G}}_{j}\left(x\right)$ are plotted, respectively, as a function of the thickness *L*_5_ of the cathode (Al) and as stated above no oscillations are found in ${\dot{G}}_{j}$ (Figure 6g) for *L*_5_ > 55 nm. This is because all the unabsorbed light from the active layer gets reflected from the interface between MoO_3_ and Al layer back into the active layer without any interference effect occurring within the Al layer. Therefore, ${\dot{G}}_{j}$ remains unaffected by any variation in the thickness of Al layer when *L*_5_ > 55 nm. For 0 < *L*_5_ < 55 nm, the reflection from Al starts to increase from zero, which increases the absorption in the active layer and hence ${\dot{G}}_{j}$ starts increasing linearly from its non-zero value at *L*_5_ = 0 nm. Such a dependence of ${\dot{G}}_{j}$ on *L*_5_ is observed for both the active layer thicknesses 95 and 180 nm (see Figure 6g). For *L*_5_ > 16 nm, the ${\dot{G}}_{j}$ in the active layer of thickness *L_j_* = 95 nm shown in Figure 6g is larger than that in *L_j_* = 180 nm. As *J_sc_* is proportional to the net ${\dot{G}}_{j}^{col}$ (see Equation (10)), this is consistent with the larger *J_sc_* in *L_j_* = 95 nm than in *L_j_* = 180 nm as illustrated in Figure 4. However, for *L*_5_ < 16 nm, this trend reverses because the photon absorption and exciton generation in the active layer increase with the thickness and the absorption of reflected photons from Al becomes negligible at such a low thickness. In view of the discussion presented above, it may be concluded that L_5_ may be equal to 55 nm or larger for obtaining maximum ${\dot{G}}_{j}$ in the active layer, which agrees very well with the *L*_5_ = 100 nm used experimentally [47]. The thickness *L*_5_ > 55 nm used in the fabrication [47] may serve the purpose of enhancing conductivity and thereby reducing the resistive power loss in Al layer.

We will now discuss the simulated results of OSC2 of the structure: Glass/ITO/PEDOT:PSS/PBDB-T-SF:IT4F/PFN-Br/Al. The simulated *J_sc_* of OSC2 is shown in Figure 4 along with that of OSC1 for comparison. It is found that *J_sc_* in OSC2 is maximum at the active layer thickness *L_j_* = 100 nm, which agrees very well with the experimental value [49]. It is also evident from Figure 4 that *J_sc_* of the conventional structure OSC2 is slightly lower than that of the inverted structure OSC1 which agrees with our previous simulation work [25], where it has been found that for a fullerene based BHJ OSC with P3HT:PCBM as the active layer, the inverted structure has a higher *J_sc_* than the conventional structure. In this view, it may be concluded that the inverted structure has higher *J_sc_* than conventional structure in both fullerene and non-fullerene acceptor based BHJ OSCs.

Further analysis of the performance of OSC2 has been carried out by simulating ${\dot{G}}_{j}$ and ${\dot{G}}_{j}\left(x\right)$ using the $n_{l}$ and $k_{l}$ values for all layers decribed above and other required parameters from Table 1. The variation of ${\dot{G}}_{j}$ with *L*_1_, *L*_4_, and *L*_5_ for OSC2 is given in Appendix A. The variation of ${\dot{G}}_{j}$ in OSC2, shown in Appendix A, with the thicknesses *L*_1_, *L*_4_, and *L*_5_ is similar to that in OSC1 shown in Figure 6, with the only difference being the magnitude of ${\dot{G}}_{j}$; in OSC2 ${\dot{G}}_{j}$ is slightly lower than that in OSC1. According to Appendix A, the optimal thickness *L*_1_ of layer 1, *L*_4_ of layer 4, and *L*_5_ of layer 5 are 148 nm, 5 nm, and 100 nm, respectively, for OSC2. However, as discussed below, the dependence of ${\dot{G}}_{j}$ and ${\dot{G}}_{j}\left(x\right)$ on the thickness *L*_2_ of PEDOT:PSS in OSC2 shown in Figure 7a,b is slightly different than that of *L*_2_ which is ZnO in OSC1 shown in Figure 6c,d.

${\dot{G}}_{j}$ plotted as a function of *L*_2_ for two active layer thicknesses $L_{j}$ = 100 nm and 180 nm in OSC2 is shown in Figure 7a and it decreases as L_2_ increases with relatively small interference effect compared to that in OSC1. According to Figure 7a, ${\dot{G}}_{j}$ at the active layer thickness $L_{j}$ = 100 nm decreases from about 1.42 *×* 10^21^ m^−^^2^s^−^^1^ at *L*_2_ = 0 nm to about 1.18 *×* 10^21^ m^−^^2^s^−^^1^ at *L*_2_ = 300 nm and for $L_{j}$ = 180 nm, it first decreases from about 1.36 *×* 10^21^ m^−^^2^s^−^^1^ at *L*_2_ = 0 nm to 1.26 *×* 10^21^ m^−^^2^s^−^^1^ at *L*_2_ = 80 nm and then increases to a maximum of 1.28 × 10^21^ m^−^^2^s^−^^1^ at *L*_2_ = 148 nm and then finally decreases again to 1.15 *×* 10^21^ m^−^^2^s^−^^1^ at *L*_2_ = 300 nm. This means that the ${\dot{G}}_{j}$ for $L_{j}$ = 180 nm in OSC2 is high for 0 nm < *L*_2_ < 30 nm unlike the case of OSC1. Although ${\dot{G}}_{j}$ is maximum at *L*_2_ = 0 nm, which is not suitable, the optimal thickness *L*_2_ of layer 2 is chosen to be 30 nm, to be consistent with the fabricated thickness [49] and with that used above for OSC1. It may be emphasized that ${\dot{G}}_{j}$ in Figure 7a at *L*_2_ = 148 nm is higher for the active layer thickness 180 nm than that for 100 nm but it is still lower than that at *L*_2_ = 30 nm for the active layer thickness of 100 nm and hence the optimal thicknesses of both layer 2 and active layer for OSC2 are chosen to be 30 nm and 100 nm, respectively.

For comparing ${\dot{G}}_{j}$ in OSC1 and OSC2, we find that at the optimal active layer thickness *L_j_* = 95 nm in OSC1, ${\dot{G}}_{j}$ at *L*_2_ = 30 nm is 1.385 × 10^21^ m^−^^2^s^−^^1^ and in OSC2 it is 1.355 × 10^21^ m^−^^2^s^−^^1^ at the optimal active layer thickness *L_j_* = 100 nm. The lower ${\dot{G}}_{j}$ in OSC2 contributes to a lower *J_sc_* which is evident from Figure 4. The contour plots of $\left|E_{j}\left(x,\lambda \right)\right|$ and ${\dot{G}}_{j}\left(x,\lambda \right)$ for OSC2 with the thicknesses of all layers optimised are plotted in Appendix A, which are apparently similar to those of OSC1 shown in Figure 5c,d. For a comparative analysis we plotted ${\dot{G}}_{j}\left(x\right)$ as a function of the position *x* within the active layer for the optimal structures of both OSC1 and OSC2 in Figure 8. The ${\dot{G}}_{j}\left(x\right)$ of OSC1 (see Figure 8) is found to be slightly higher than that of OSC2 at the position *x* = 0 nm and the difference reduces gradually towards the end of the active layer thickness *L_j_* and hence the inverted OSC1 can be expected to show better performance compared to conventional OSC2. As the optimised thickness *L_j_* of the active layer of OSC1 is 95 nm and that of OSC2 is 100 nm, the amount of active layer material needed for the inverted OSC1 is less than that of the conventional OSC2 which may lead to lower fabrication cost in the mass production of inverted structure compared to the conventional structure.

Finally, we discuss the simulation results of OSC3, which is a fullerene acceptor based BHJ OSC of the conventional structure. *J_sc_* of OSC3 plotted in Figure 4, reveals the optimal active layer thickness of 80 nm, which is in reasonable agreement with the experimental thickness of 70 nm [50,51]. It is also very conclusive from Figure 4 that OSC3 produces relatively the lowest *J_sc_* in comparison with OSC1 and OSC2, which may be attributed to the use of fullerene acceptor [60]. Following the procedures for OSC1 and OSC2, ${\dot{G}}_{j}$ and ${\dot{G}}_{j}\left(x\right)$ as a function of the thickness of layer 1, layer 2, layer 4, and layer 5 for OSC3 are also simulated using the $n_{l}$ and $k_{l}$ values for P3HT:PCBM from [50,51,53,54,55], for the Glass, ITO, PEDOT:PSS, LiF and Al from [50,51,53,54,55] and other required parameters from Table 1 in Equations (1) and (2), respectively. ${\dot{G}}_{j}$ (Equation (1)) is calculated for two active layer thickness *L_j_* = 80 nm and 180 nm as a function of the thicknesses of other layers whereas the corresponding contour plots of ${\dot{G}}_{j}\left(x\right)$ (Equation (2)) is calculated as a function of *x* within the active layer only of the optimal thicknesses *L_j_* = 80 nm. The variations in ${\dot{G}}_{j}$ and ${\dot{G}}_{j}\left(x\right)$ of OSC3 with respect to the thickness of layer 4 and layer 5 (see Appendix A) show similar trends as those in OSC1 (see Figure 6) and OSC2 (see Appendix A) and their simulated optimal thickness are obtained as 1 nm and 100 nm, respectively. The variations in ${\dot{G}}_{j}$ and ${\dot{G}}_{j}\left(x\right)$ with the thickness of layers 1 and 2 are shown in Figure 9a–d.

The variation of ${\dot{G}}_{j}$ with *L*_1_ for the active layer thicknesses of *L_j_* = 80 nm and 180 nm in OSC3, as shown in Figure 9a, exhibits larger oscillatory nature compared to OSC1 (Figure 6a) and OSC2 (Appendix A). This may be attributed to the larger interference effect caused by the combination of fullerene-based accepter in the active layer and layer 1. In view of this, a small variation from the optimal thickness of the layer 1 may lead to a larger loss in ${\dot{G}}_{j}$, and hence in *J_sc_* in the fullerene based OSC3 in comparison with that in non-fullerene based OSC1 and OSC2. From Figure 9a, the optimal thickness *L*_1_ of layer 1 is found to be 46 nm at the optimal active layer thickness *L_j_* = 80 nm. However, the thickness of ITO used in the fabrication of fullerene based OSC3 is usually 150–180 nm [50,51,68], which corresponds to the second maximum in Figure 9a and it is lower than the first one at 46 nm. Although the optimal thickness of 46 nm determined here from the simulation agrees with 40 nm used in our earlier simulation work [25], a thicker ITO layer used in the fabrication might help reducing the resistive power loss in the series resistance of layer 1 because ITO sheet resistance decreases significantly with the increase in its thickness [61]. It may be noted that, the first and second maxima shown in Figure 9a at the active layer thickness of 80 nm, are also clearly visible as widest bright spots at *L*_1_ = 46 nm and 180 nm, respectively, in Figure 9b.

In Figure 9c, we plotted ${\dot{G}}_{j}$ as a function of the thickness *L*_2_ of PEDOT:PSS in OSC3 for the two active layer thicknesses *L_j_* = 80 nm and 180 nm and the contour plot of ${\dot{G}}_{j}\left(x\right)$ as a function of the thickness *L*_2_ and position *x* within the active layer of optimal thickness $L_{j}$ = 80 nm is shown in Figure 9d. According to Figure 9c, the ${\dot{G}}_{j}$ is maximum at *L*_2_ = 0 nm for both the active layer thicknesses and hence the maximum exciton generation rate may occur in the absence of layer 2. This is also evident from Figure 9d, where the widest bright spot appears at *L*_2_ = 0 nm. This trend of maximum ${\dot{G}}_{j}$ at *L*_2_ = 0, shown in Figure 9c is similar to that found in OSC2 (see Figure 7a) and also in OSC1 for *L_j_* = 95 nm but for *L_j_* = 180 nm the maximum ${\dot{G}}_{j}$ occurs at non-zero thickness *L*_2_ (see Figure 6c). As mentioned above, the layer 2 is used to improve the charge carrier extraction efficiency and hence it cannot be of zero thickness. In the experimentally fabricated OSC3 [50,51], a thickness of *L*_2_ = 45 nm was used, and according to Figure 9c, the simulated ${\dot{G}}_{j}$ was found to be 6.73 × 10^20^ m^−2^s^−1^. Although a similar ${\dot{G}}_{j}$ may be found for *L*_2_ = 100 nm (see Figure 9c), the thinner layer may be preferred for reducing the amount of material used. Figure 9c also reveals that ${\dot{G}}_{j}$ for the active layer thickness of 180 nm is higher than that for 80 nm in the whole range of thickness *L*_2_ but the optimal thickness of 80 nm is preferred. This may be due to the fact that the fill factor of OSCs decreases almost linearly as the thickness of active layer increases [42,59,69], which is caused by the enhanced recombination [59,69]. The contour plots of $\left|E_{j}\left(x,\lambda \right)\right|$ and ${\dot{G}}_{j}\left(x,\lambda \right)$ for OSC3 with the thicknesses of all layers optimised are plotted in Appendix A.

The simulation presented in this paper is expected to help in optimising the layered structure of both non-fullerene and fullerene based BHJ OSCs and has revealed that the inverted NF BHJ OSC1 has relatively better *J_sc_* than the conventional NF BHJ OSC2 and fullerene based OSC3. The optimised structures thus simulated for all three OSCs are given in Table 2.

## 4. Conclusions

Characterisation of the exciton generation rates has been carried out in three different BHJ OSCs: (i) an inverted non-fullerene acceptor based OSC1; (ii) a conventional non-fullerene acceptor based OSC2; and (iii) a conventional fullerene acceptor based OSC3. It was found that the regions of overlap of the constructive interference of forward travelling and reflected light waves and those of the high photon absorption regions depend on the thickness of active layer which is used to optimise the thickness of the active layer with the maximum overlap. The rates of total exciton generation ${\dot{G}}_{j}$ and position dependent exciton generation ${\dot{G}}_{j}\left(x\right)$ in the active layer have been optimised with respect to the thicknesses of each layer in the device structure. The optimal simulated designs thus obtained for the three OSCs are given in Table 2. It is expected that the results of this paper may help in fabricating highly efficient and cost-effective BHJ OSCs.

## Figures and Tables

**Figure 1 nanomaterials-11-00209-f001:**
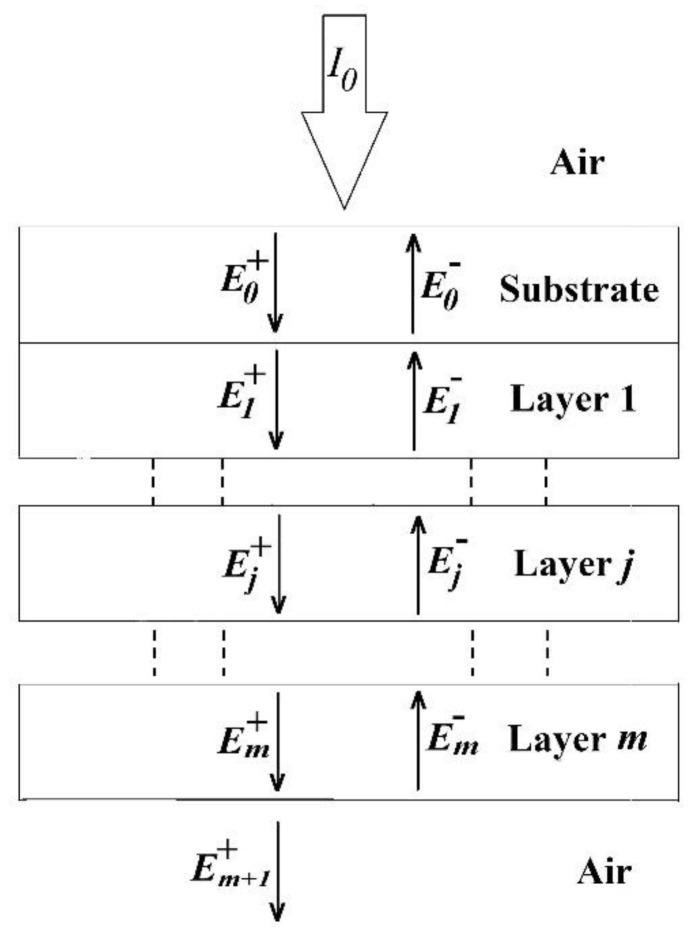
Schematic diagram of a bulk-heterojunction organic solar cell (BHJ OSC) as a multilayer stack suitable for the optical transfer matrix method (OTMM). The electric field components of the incident and reflected electromagnetic waves in the *l*^th^ layer are represented by $E_{l}^{+}$ and $E_{l}^{-}$, respectively, where l=0,1,…j…,m+1 and *j* denotes the active layer.

**Figure 2 nanomaterials-11-00209-f002:**
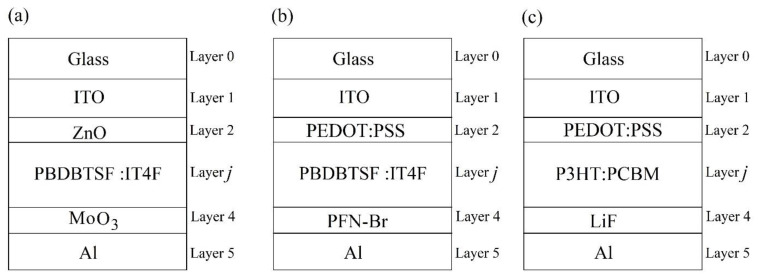
Three BHJ OSC structures considered in the simulation: (**a**) OSC1—an inverted BHJ OSC with non-fullerene acceptor, (**b**) OSC2—a conventional BHJ OSC with non-fullerene acceptor, and (**c**) OSC3—a conventional BHJ OSC with fullerene acceptor. Layer 1 is the front electrode (FE), layer 2 is the front charge transport layer (FCTL), layer *j* is the active layer, layer 4 is the rear charge transport layer (RCTL), and layer 5 is the rear electrode (RE).

**Figure 3 nanomaterials-11-00209-f003:**
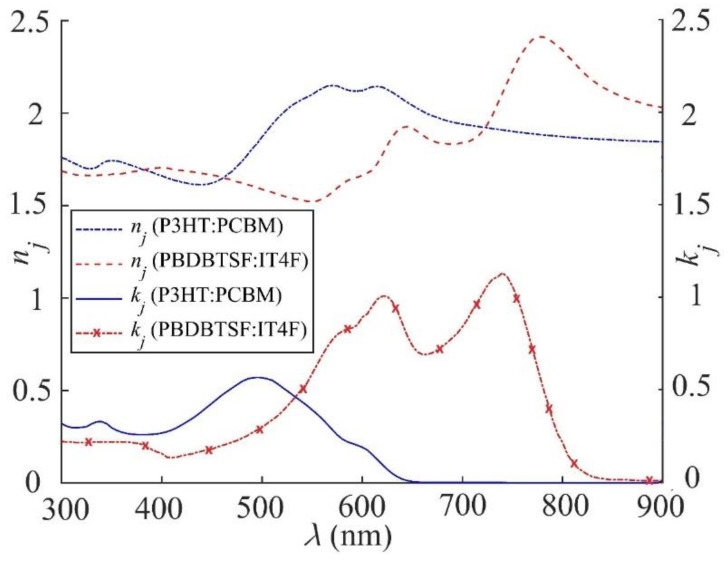
The experimental values of the refractive index (*n_j_*) and extinction coefficient (*k_j_*) of the active layer materials P3HT:PCBM [51,55] and PBDBTSF:IT4F [49,52] plotted as a function of the wavelength of the incident light.

**Figure 4 nanomaterials-11-00209-f004:**
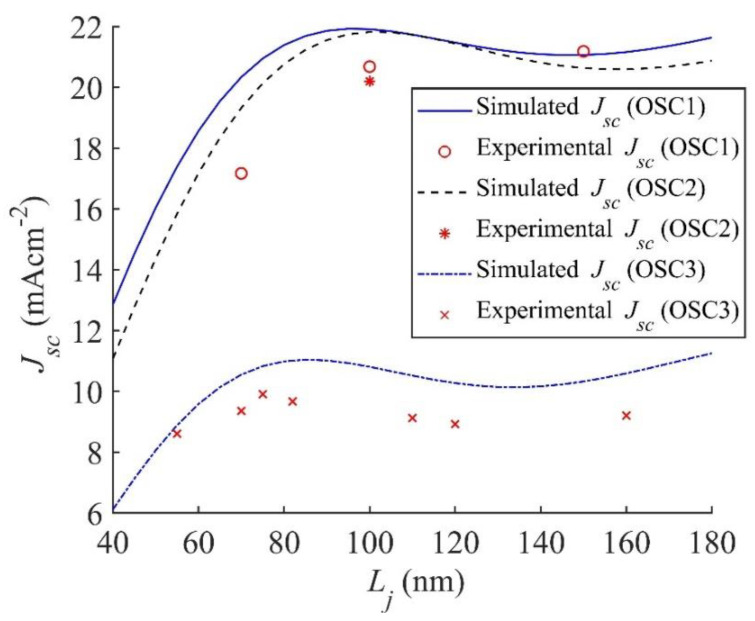
Simulated *J_sc_* of OSC1 of the structure Glass/ITO/ZnO/PBDB-T-SF:IT4F/MoO_3_/Al, OSC2 of the structure Glass/ITO/PEDOT:PSS/PBDB-T-SF:IT4F/PFN-Br/Al and OSC3 of the structure Glass/ITO/PEDOT:PSS/P3HT:PCBM/LiF/Al are plotted as a function of the thickness of the active layer along with their experimental results from [47,49,50,51], respectively.

**Figure 5 nanomaterials-11-00209-f005:**
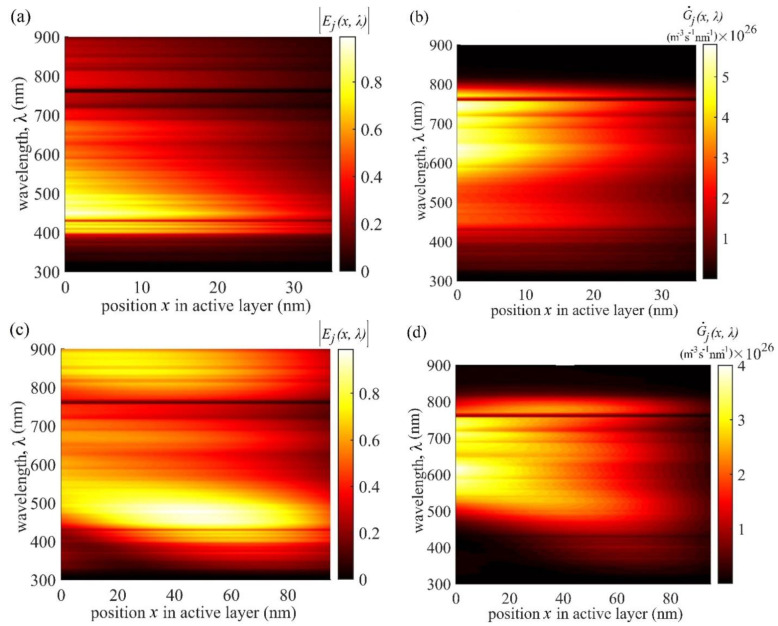
Left column presents the contour plots of $\left|E_{j}\left(x,\lambda \right)\right|$ from Equation (5) in the active layer of OSC1 for three different thicknesses $L_{j}$, (**a**) 35 nm, (**c**) 95 nm, and (**e**) 180 nm. The right column presents the corresponding ${\dot{G}}_{j}\left(x,\lambda \right)$ contour plots from Equation (3) for the three thicknesses: (**b**) 35 nm, (**d**) 95 nm, and (**f**) 180 nm. The brighter spots on the left figures represent regions of the constructive interference of electric field of electromagnetic (EM) waves within the active layer and the darker spots represent regions of destructive interference. Likewise, the brighter spots on the right figures represent the positions of higher ${\dot{G}}_{j}\left(x,\lambda \right)$ and darker spots that of lower ${\dot{G}}_{j}\left(x,\lambda \right).$

**Figure 6 nanomaterials-11-00209-f006:**
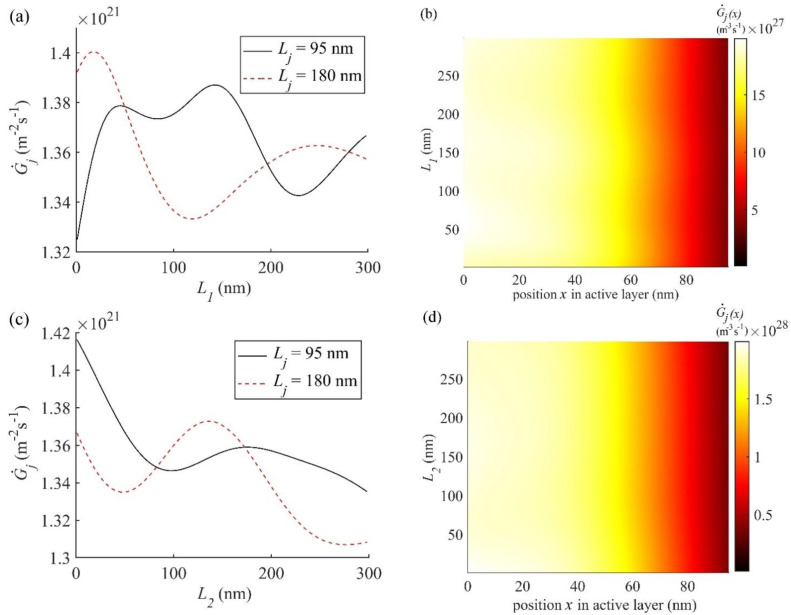
(Left column) ${\dot{G}}_{j}$ is plotted for two active layer thicknesses $L_{j}$ = 95 nm and 180 nm, as function of the thickness of (**a**) *L*_1_ of layer 1 (ITO), (**c**) *L*_2_ of layer 2 (ZnO), (**e**) *L*_4_ of layer 4 (MoO_3_), and (**g**) *L*_5_ of layer 5 (Al). (Right column) Contour plots of ${\dot{G}}_{j}\left(x\right)$ as a function *x* within the active layer of optimal thickness $L_{j}$ = 95 nm, for other layers of thickness (**b**) *L*_1_ of layer 1 (ITO), (**d**) *L*_2_ of layer 2 (ZnO), (**f**) *L*_4_ of layer 4 (MoO_3_), and (**h**) *L*_5_ of layer 5 (Al).

**Figure 7 nanomaterials-11-00209-f007:**
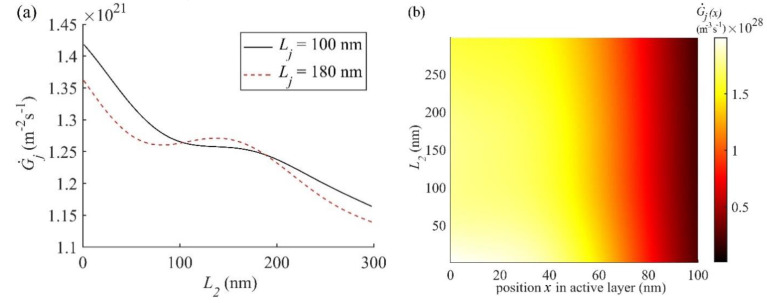
(**a**) ${\dot{G}}_{j}$ is plotted as function of the thickness *L*_2_ of layer 2 (PEDOT:PSS) in OSC2 for two active layer thicknesses $L_{j}$ = 100 nm and 180 nm. (**b**) Contour plots of ${\dot{G}}_{j}\left(x\right)$ as a function of thickness *L*_2_ of layer 2 and position *x* within the active layer of OSC2 at the optimal thickness $L_{j}$ = 100 nm of the active layer.

**Figure 8 nanomaterials-11-00209-f008:**
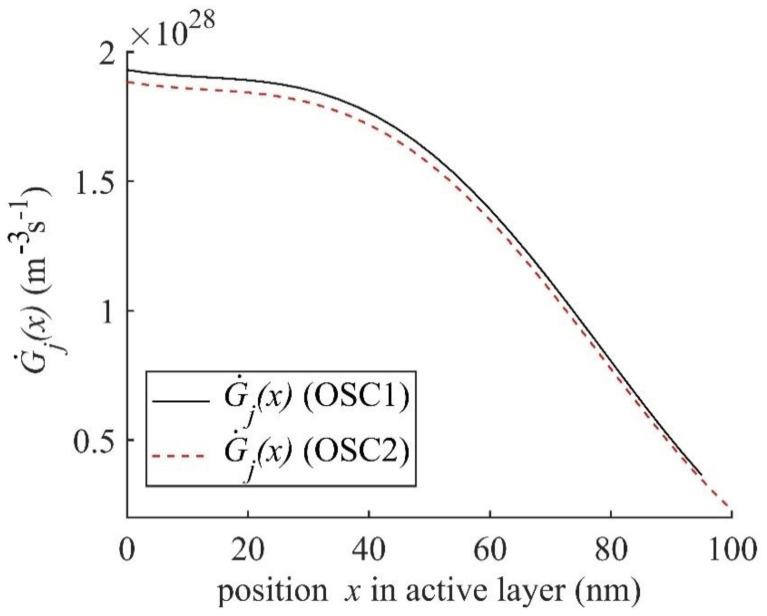
${\dot{G}}_{j}\left(x\right)$ plotted as a function of the position *x* within the active layer of optimal thickness *L_j_* = 95 nm and 100 nm for OSC1 and OSC2, respectively.

**Figure 9 nanomaterials-11-00209-f009:**
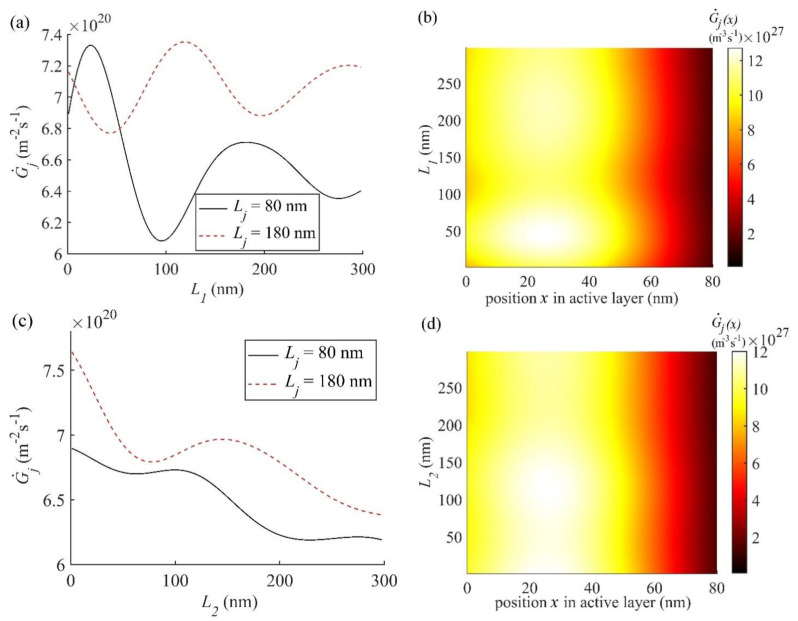
(**a**) ${\dot{G}}_{j}$ is plotted as function of the thickness *L*_1_ of layer 1 (ITO) for two active layer thicknesses $L_{j}$ = 80 nm and 180 nm, (**b**) contour plot of ${\dot{G}}_{j}\left(x\right)$ as function of the thickness *L*_1_ of ITO and position *x* within the active layer of optimal thickness $L_{j}$ = 80 nm, (**c**) ${\dot{G}}_{j}$ is plotted as function of the thickness *L*_2_ of layer 2 (PEDOT:PSS) for two active layer thicknesses $L_{j}$ = 80 nm and 180 nm, and (**d**) contour plot of ${\dot{G}}_{j}\left(x\right)$ as function of the thickness *L*_2_ of PEDOT:PSS and position *x* within the active layer of optimal thickness $L_{j}$ = 80 nm.

**Table 1 nanomaterials-11-00209-t001:** Input parameters required in the simulation.

	P3HT:PCBM	PBDB-T-SF:IT-4F
$L_{j}$ (nm)	40–180	40–180
$\varepsilon_{r}$	3.0 [57]	3.40 [58]
$\mu_{n}$ (m^2^V^−1^s^−1^)	3 × 10^−7^ [59]	6.27 × 10^−9^ [58]
$\mu_{p}$(m^2^V^−1^s^−1^)	3 × 10^−8^ [59]	2.26 × 10^−8^ [58]
ξ	1.3 × 10^−3^ [60]	1.6 × 10^−3^ [58]
$R_{C}$ (${\Omega \mathrm{cm}}^{2}$)	3.0 [60]	1.0 [60]
ρ (Ωcm)	4000 [60]	1000 [60]
$R_{sheetITO}$(Ω/sq)	18.0 [61]	13.7 [41,60]
$L_{ITO}$ (cm)	0.5 [50]	1 [47]
*n*	3.2 × 10^22^ [62]	
λ1 (nm)	300	
λ2 (nm)	900	
$N_{t.bulk}$$(m^{-3}$)	2.1 × 10^19^ [22]	
$N_{t.surf}(m^{-2}$)	6.3 × 10^18^ [22]	
*T* (K)	300 [25]	
*e* (C)	1.6 × 10^−19^ [63]	
h (m^2^kgs^−1^)	6.63 × 10^−34^ [64]	
$\varepsilon_{0}$ (Fm^−1^)	8.85 × 10^−12^ [65]	
*k* (m^2^kgs^−2^K^−1^)	1.38 × 10^−23^ [66]	

**Table 2 nanomaterials-11-00209-t002:** Optimised layer thickness for all three OSCs.

	OSC1	OSC2	OSC3
*L*_1_ (nm)	148 (ITO)	148 (ITO)	180 (ITO)
*L*_2_ (nm)	30 (ZnO)	30 (PEDOT:PSS)	45 (PEDOT:PSS)
*L_j_* (nm)	95 (PBDBTSF:IT4F)	100 (PBDBTSF:IT4F)	80 (P3HT:PCBM)
*L*_4_ (nm)	10 (MoO_3_)	5 (PFN-Br)	1 (LiF)
*L*_5_ (nm)	100 (Al)	100 (Al)	100 (Al)

## Data Availability

The data presented in this study are available in this article and Appendix A.

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
