# Peer review of "Characterising Exciton Generation in Bulk-Heterojunction Organic Solar Cells"

_nanomaterials, 2021, doi:10.3390/nano11010209_

Round 1
Reviewer 1 Report
In this paper, exciton generation for three bulk heterojunction solar cell architectures is simulated and the effect of thickness variation of all involved layers is investigated.
The authors properly explain their model, so the results can be easily comprehended by the reader. It is interesting to see how well the simulations match experimental data.
The paper can give valuable guidance to the experimentalist, especially when pointing to which thicknesses have the largest influence on performance.
It would be helpful, if the authors include the extinction coefficients of the active layer materials in a figure, so the different behavior can be immediately understood.
To find the optimal layer thicknesses, the authors only vary the thickness of one layer, while leaving the other values constant. Can the authors comment on the significance of cross-correlations?
Author Response
We would like to thank both the reviewers for their constructive comments. All the changes made in the revised manuscript are highlighted.
Reviewer 1:
Comment 1: It would be helpful, if the authors include the extinction coefficients of the active layer materials in a figure, so the different behaviour can be immediately understood.
Response: The refractive index nj extinction coefficient kj of the active layer materials are plotted in Fig. 3, a new figure added in the revised manuscript on page 7 (line 3) and all other figures have been re-numbered as highlighted throughout the revised manuscript.
Comment 2: To find the optimal layer thicknesses, the authors only vary the thickness of one layer, while leaving the other values constant. Can the authors comment on the significance of cross-correlations?
Response 2: Thank you for this comment but such a correlation is nearly impossible to make. You cannot change more than one thickness at the same time for optimising the property of a layered structure.
Reviewer 2 Report
In this work by Ram et al., the authors simulated exciton generation in bulk-heterojunction organic solar cells. This is a computation study, which would gain very much if the calculated results were verified experimentally as the proof of concept is quite interesting. To compensate for the lack of it, the authors presented a thorough explanation on this topic from modeling perspective. Overall, I have no major objections to the reported research. There are only minor improvements, which could be implemented to make this contribution fit the Nanomaterials journal.
1) Abstract is a very important part of the article as it is decisive for the readers whether or not to read the article. However, the abstract in the current state is quite unclear and confusing e.g. "It is found that the overlap of the regions of strong constructive interference of incident and reflected electric fields of electromagnetic waves depends on the active layer thickness which is used to optimise the thickness of active layer at which the overlap becomes maximum" Thickness of active layer optimizes the thickness of the active layer? Please carefully read the abstract and correct it to make it more approachable.
2) One of the necessary conditions to make a work publishable is to include all the details, which can enable others to reproduce it. Only then, others can build on these findings. However, this article lacks some precision on this front. More details should be given regarding the kind of software and methods employed for the article.
3) The article actually is quite long and thus following the narration is challenging. The authors already decided to include a SI file, which is good, but more information should be moved into the supplement from the main text.
4) Formatting of the plots could be improved to make the results more legible. Please consider increasing the font size in the plots.
Author Response
We would like to thank both the reviewers for their constructive comments. All the changes made in the revised manuscript are highlighted.
Reviewer 2:
Comment 1: Abstract is a very important part of the article as it is decisive for the readers whether or not to read the article. However, the abstract in the current state is quite unclear and confusing e.g. "It is found that the overlap of the regions of strong constructive interference of incident and reflected electric fields of electromagnetic waves depends on the active layer thickness which is used to optimise the thickness of active layer at which the overlap becomes maximum" Thickness of active layer optimizes the thickness of the active layer? Please carefully read the abstract and correct it to make it more approachable.
Response 1: Thank you for pointing out this typographical error. We have modified that sentence in the abstract of the revised manuscript as, “It is found that the overlap of the regions of strong constructive interference of incident and reflected electric fields of electromagnetic waves and those of high photon absorption within the active layer depends on the active layer thickness. An optimal thickness of the active layer can thus be obtained at which this overlap is maximum.”
Comment 2: One of the necessary conditions to make a work publishable is to include all the details, which can enable others to reproduce it. Only then, others can build on these findings. However, this article lacks some precision on this front. More details should be given regarding the kind of software and methods employed for the article.
Response 2: No commercial software has been used in this work. We have programmed OTMM ourselves and used it on MATLAB for the simulation. We have added a sentence on page 4 (lines 1-2) above Figure 1 in the revised manuscript in this regard. This is the reason for comparing our results first with the experimental ones in Fig. 4.
Comment 3: The article actually is quite long and thus following the narration is challenging. The authors already decided to include a SI file, which is good, but more information should be moved into the supplement from the main text.
Response 3: In the presentation of the manuscript, authors believe that all information in the article are important for readers and therefore we have not moved into supplementary file as suggested by the reviewer. Such a separation may further increase the difficulty in following the narration.
Comment 4: Formatting of the plots could be improved to make the results more legible. Please consider increasing the font size in the plots.
Response 4: The font size in all Figs 3, 4, 5 (a-f), 6(a-h), 7 (a,b), 8, 9(a-d), S1(a-c), S2 (a,b), S3(a,b) and S4(a,b) have been increased. Thank you for the suggestion.